# Circulating tRNA-Derived Small RNAs as Novel Radiation Biomarkers of Heavy Ion, Proton and X-ray Exposure

**DOI:** 10.3390/ijms222413476

**Published:** 2021-12-15

**Authors:** Wenjun Wei, Hao Bai, Yaxiong Chen, Tongshan Zhang, Yanan Zhang, Junrui Hua, Jinpeng He, Nan Ding, Heng Zhou, Jufang Wang

**Affiliations:** 1Key Laboratory of Space Radiobiology of Gansu Province & CAS Key Laboratory of Heavy Ion Radiation Biology and Medicine, Institute of Modern Physics, Chinese Academy of Sciences, Lanzhou 730000, China; weiwenjun@impcas.ac.cn (W.W.); baihao@impcas.ac.cn (H.B.); chenyx07@impcas.ac.cn (Y.C.); zhangtongshan@impcas.ac.cn (T.Z.); zhangyanan@impcas.ac.cn (Y.Z.); huajunrui@impcas.ac.cn (J.H.); hejp03@impcas.ac.cn (J.H.); dn@impcas.ac.cn (N.D.); hengzhou@impcas.ac.cn (H.Z.); 2Institute of Modern Physics, University of Chinese Academy of Sciences, Beijing 100049, China

**Keywords:** tsRNA, serum, biomarkers, radiation exposure, multi-factor models

## Abstract

The effective and minimally invasive radiation biomarkers are valuable for exposure scenarios in nuclear accidents or space missions. Recent studies have opened the new sight of circulating small non-coding RNA (sncRNA) as radiation biomarkers. The tRNA-derived small RNA (tsRNA) is a new class of sncRNA. It is more abundant than other kinds of sncRNAs in extracellular vesicles or blood, presenting great potential as promising biomarkers. However, the circulating tsRNAs in response to ionizing radiation have not been reported. In this research, Kunming mice were total-body exposed to 0.05–2 Gy of carbon ions, protons, or X-rays, and the RNA sequencing was performed to profile the expression of sncRNAs in serum. After conditional screening and validation, we firstly identified 5 tsRNAs including 4 tRNA-related fragments (tRFs) and 1 tRNA half (tiRNA) which showed a significant level decrease after exposure to three kinds of radiations. Moreover, the radiation responses of these 5 serum tsRNAs were reproduced in other mouse strains, and the sequences of them could be detected in serum of humans. Furthermore, we developed multi-factor models based on tsRNA biomarkers to indicate the degree of radiation exposure with high sensitivity and specificity. These findings suggest that the circulating tsRNAs can serve as new minimally invasive biomarkers and can make a triage or dose assessment from blood sample collection within 4 h in exposure scenarios.

## 1. Introduction

Ionizing radiation from nuclear accidents, outer space, or radiation facilities is a major concern for human health. The widely used physical dosimeters have limitations in indicating the individual responses because the biological effects may be different between different individuals [1]. Radiation-responsive biomarkers can indicate the individual absorbed dose and provide information about radiation-induced biological damage. Thus, the promising radiation biomarkers are valuable for personalized assessment in exposure scenarios, especially for low doses of radiation exposure which cause mild symptoms or cancer risk by a progressive pattern. The current radiation biomarkers mainly based on DNA damage such as chromosomal aberration and micronuclei formation, have been developed for many years [2,3,4]. However, the detection processes of these biomarkers are time-consuming and complex [5,6]. These considerable limitations make an urgent demand for new rapid detection and minimally invasive biomarkers.

Extracellular RNAs (exRNAs) can stably exist in blood and are resistant to hydrolysis in the process of extraction and preservation [7]. This property makes them attractive candidates of minimally invasive biomarkers for indicating cancers or diseases [8,9]. exRNAs are enriched in various small non-coding RNAs (sncRNAs) such as microRNA (miRNA), tRNA and Y RNA fragments, piRNA and small nuclear RNA (snRNA), etc. [10]. Recent studies have reported the discovery of miRNAs in serum or plasma as biomarkers of radiation exposure [11,12,13]. A study on RNA composition of exRNAs showed that tRNA-derived small RNA (tsRNA) is more enriched than miRNA and other kinds of sncRNAs [14], indicating tsRNA in blood is more easily and accurately detected. The tsRNA is a new class of sncRNA generated through cleavage of mature tRNAs, which can be broadly classified into two main groups: tRNA related fragments (tRFs) generated from mature or precursor tRNAs, and tRNA halves (tiRNAs) generated by specific cleavage in the anticodon loops of mature tRNAs [15]. The tsRNA was once thought to be a meaningless degradation byproduct of tRNA homoeostasis. However, high-throughput sequencing techniques fostered the discovery of a variety of tsRNAs in different samples or pathological and stress conditions, uncovering the regulatory world of tsRNA beyond canonical tRNA biology [16]. Increasing evidence indicates that tsRNAs participate in transcriptional and post-transcriptional gene expression by targeting 3ʹ-untranslated regions (UTRs) of mRNAs just like miRNA does. Moreover, a recent study showed that there are putative interactions between tsRNAs and noncoding RNAs and mRNA introns [17]. Except for the translational RNAi machinery, tsRNAs can bind ribosome or translation factors directly to interfere with their functions [18]. These regulatory roles make tsRNAs closely associated with diseases or physical disorders [19,20,21]. While, Fricker R et al. discovered the tsRNA signatures in cells exposed to different stress conditions [22], suggesting the possibility of tsRNAs as biomarkers for environmental stressors such as ionizing radiation. However, the circulating tsRNAs in response to ionizing radiation have not been reported.

Inspired by the above studies, we expect that circulating tsRNAs in serum could serve as new biomarkers for radiation exposure. Here, mice were total-body exposed to different radiations including carbon ions, protons, and X-rays. The differentially expressed tsRNAs in serum were screened by RNA sequencing and further validated to observe their dose- and time-responses after exposure. Furthermore, we tried to develop a method to indicate the degree of radiation exposure using the tsRNA. We sought to expand current studies in circulating tsRNA and explore its potential of serving as a radiation biomarker.

## 2. Results

### 2.1. Profiling of tsRNA in Serum after Total-Body Exposure of Mice to Carbon Ions

To profile the tsRNA in serum after heavy ion exposure, Kunming mice were total-body irradiated by 0.05, 0.1, 0.5, 1 Gy of carbon ions, and non-irradiated mice (0 Gy) were set as the control group (*n* = 3/dose), and serum samples were collected at 24 h after exposure. Then, the expression of serum sncRNAs was analyzed by RNA sequencing. Only the high abundance sequencing reads within the length of 14–40 nucleotides (nt) were recorded and mapped to tRNA database of mice. We analyzed the length distribution of reads counts against the lengths of reads in different dose groups. There were two evident peaks of the read counts at 19–24 nt and 28–32 nt in irradiation groups and the control group. Notably, the majority of reads were distributed around 30 nt (Figure 1A). Through annotation with tRNA and miRNA database, we classified these sncRNAs, and the statistical information is shown in Appendix A. Over 75% of sncRNA reads were from the mature tRNAs, 1% of reads were from pre-tRNAs or miRNAs in all dose groups (Figure 1B). The fragments from the mature tRNAs were analyzed for observing their subtype of tRF or tiRNA. It was found that most of tsRNAs were classified as tRF-5c, while only 3–6% of the fragments were classified as tiRNA (Figure 1C). These results suggest that the tRF-5c and tiRNA from the 5′-end of mature tRNAs have a high abundance in mice serum.

### 2.2. Screening and Validation of Differentially Expressed Serum tsRNAs after Carbon Ion Irradiation

To further verify the differentially expressed tsRNAs from expression profiles, the tsRNAs with the relative expression changes higher than 1.5 times (compared with the control group) were screened (Figure 2A). Among them, the levels of 17 tsRNAs increased and 9 tsRNAs decreased in all irradiation groups (Figure 2B). Subsequently, we chose 10 tsRNAs with higher read counts for validation, including 4 upregulated tsRNAs and 6 downregulated tsRNA (Appendix A). Mice were total-body irradiated by 0.05, 0.1, 0.5, 1 Gy of carbon ions and compared to non-irradiated mice (*n* = 6/dose). The relative expression of 10 selected serum tsRNAs at 24 h post-irradiation was detected by RT-qPCR. Their expression patterns were shown by hierarchical clustering heat map (Figure 2C), and the relative expression changes compared to the control group were shown by fold change histogram (Figure 2D). We found that the level of the 10 most modulated tsRNAs decreased after irradiation, and 5 tsRNAs showed significant decrease in at least three dose groups. Their confirmed names in the Genomic tRNA Database were tiRNA-Glu-TTC-003, tRF-Val-AAC-024, tRF-Gln-CTG-018, tRF-Lys-CTT-008, and tRF-Lys-TTT-019. Thus, these 5 tsRNAs were selected as candidate biomarkers for further validation. Their mature sequences and the designed RT-qPCR primers are shown in Appendix A.

### 2.3. The Dose Responses of Five Selected tsRNAs after Exposure of Mice to Carbon Ions, X-rays, and Protons

As shown in Figure 3A, the 5 selected tsRNAs including 4 tRF-5c and 1 tiRNA-5 are all from the 5′-end of tRNAs (Figure 3A). To further investigate the dose responses of five selected tsRNAs to different radiation types, mice were total-body irradiated by carbon ions (0, 0.05, 0.1, 0.5, and 1 Gy; *n* = 11/dose), X-rays (0, 0.1, 0.5, 1, and 2 Gy; *n* = 5/dose), or protons (0, 0.1, 0.5, 1, and 2 Gy; *n* = 4/dose), separately. The relative expression of 5 selected tsRNAs at 24 h post-irradiation was quantified by RT-qPCR. As shown in Figure 3B–F, all 5 selected tsRNAs showed a decrease trend with increasing doses after exposure of mice to carbon ions, X-rays, or protons. Notably, the levels of tiRNA-Glu-TTC-003, tRF-Val-AAC-024, and tRF-Gln-CTG-018 decreased significantly (*p* < 0.05) from 0.05 to 2 Gy in response to different radiation types. These results suggest that the 5 selected tsRNAs positively respond to carbon ion, X-ray, and proton exposure, their decrease at low doses highlighting their great advantages of being radiation biomarkers.

### 2.4. Temporal Changes of the Five Selected tsRNAs in Serum of Mice after Radiation Exposure

An early and long-term response after exposure can make the biomarker more detectable, so we explored the temporal changes of the 5 selected tsRNAs in serum of mice from 6 h to 30 days (d) after exposure. Here, mice were total-body irradiated by 1 Gy of carbon ions or 2 Gy of X-rays, and the serum samples were collected at 6 h, 24 h, 3 d, 7 d, 15 d and 30 d (*n* = 3–4/time point). The relative expression of the 5 selected tsRNAs at different time points versus the control group (non-irradiated mice) was detected by RT-qPCR. As shown in Figure 4A–E, the 5 tsRNAs showed a significant downward trend from 6 h to 7 d after carbon ion or X-ray exposure. Interestingly, the downward trend of the 5 tsRNAs became insignificant at 15 d after carbon ion exposure but persisted up to 30 d after X-ray exposure. These results demonstrate that the 5 selected serum tsRNAs can respond to radiation exposure at an early stage, and the best time window for detection is 6 h to 7 d after exposure.

### 2.5. Establishment of a Model Based on Multiple tsRNA Biomarkers to Indicate Radiation Exposure

Through validation experiments above, we identified 5 serum tsRNAs which present great potential as biomarkers of carbon ion, X-ray, and proton exposure. Further, we attempted to establish multi-factor models combining the 5 tsRNAs to indicate the exposure risk of carbon ions, X-rays, and protons, respectively. The logistic regression model was used to fit the dose–response data. As shown in the flow chart (Figure 5A), we divided different dose ranges into different exposure degrees (ED). Using these ED values as categorical variables, the dose–response data of the 5 tsRNAs that received carbon ion, X-ray, or proton irradiation were used to perform the logistic regression, respectively. It was found that a multiple linear regression gave the best fitting results, the pattern of the equation is:
Y = β_0_ + β_1_X_1_ + β_2_X_2_ + ··· + β_i_X_i_(1)

In Equation (1), Y was defined as exposure risk score (*ERS*). X was the dose–response data of tsRNAs, and β is the partial regression coefficient. The coefficients of fitting the models by SPSS are shown in Appendix A. Three models to evaluate the association between exposure risk score (*ERS*) and the relative expression of the 5 tsRNAs is:
*ERS(Carbon ions)* = *0.646* − *0.149*(*X*_*tiRNA-Glu-TTC-003*_) + *0.022*(*X*_*tRF-Val-AAC-024*_) − *0.093*(*X*_*tRF-Gln-CTG-018*_) + *0.139*(*X*_*tRF-Lys-CTT-008*_) − *0.206*(*X*_*tRF-Lys-TTT-019*_)(2)
*ERS(X-rays)* = *0.943* − *0.05*(*X*_*tRF-Val-AAC-024*_) − *0.053*(*X*_*tRF-Gln-CTG-018*_) − *0.013*(*X*_*tRF-Lys-CTT-008*_) + *0.023*(*X*_*tRF-Lys-TTT-019*_)(3)
*ERS(protons)* = *0.521* − *0.081*(*X*_*tiRNA-Glu-TTC-003*_) + *0.788*(*X*_*tRF-Val-AAC-024*_) − *0.077*(*X*_*tRF-Gln-CTG-018*_) − 
*0.646*(*X*_*tRF-Lys-CTT-008*_) − *0.077*(*X*_*tRF-Lys-TTT-019*_)(4)

After substituting the relative expression data of 5 tsRNAs into Equations (2)–(4), respectively, a set of *ERS* values was calculated in each ED. The correlation analysis showed that *ERS* values of carbon ions, X-rays, and protons had a positive linear correlation to ED (Figure 5B–D). Moreover, the receiver-operating characteristic (ROC) analysis was performed to investigate the capacity of multi-factor models or single tsRNA in predicting ED. ROC curve showed the discriminative ability of a test by the position of the full curve in a graph which indicates the relation between the true positive rate (TPR) and the false positive rate (FPR) over a wide range of cut-off points. Here, ROC curves of multi-factor models and the single tsRNA were depicted, and the area under the ROC curve (AUC) was calculated, respectively. The larger AUC value means the higher sensitivity and specificity in prediction. The ROC analysis showed that the AUC values of the models are 0.955 (for carbon ion irradiation), 0.974 (for X-ray irradiation), and 0.984 (for proton irradiation). Notably, the AUC values from multi-factor models are bigger than that from the single tsRNA (Figure 5E–G). These results indicate that the multi-factor model based on the 5 tsRNA biomarkers is effective to predict the exposure degree of different radiations with higher sensitivity and specificity.

### 2.6. The Radiation Responses of the Five tsRNA Biomarkers in Different Mice Species and Their Expression Levels in Human Serum

To further investigate the radiation responses of the 5 tsRNA biomarkers in different mouse strains, C57BL/6 and Balb/c mice were total-body irradiated by 0.1, 0.5, 1 Gy of carbon ions and compared to non-irradiated mice (0 Gy). The relative expression of the 5 tsRNAs at 24 h post-irradiation was detected by RT-qPCR. Similar to the results of Kunming mice, the relative expression of 5 tsRNAs showed a significant decrease with increasing doses in other mouse strains (Figure 6A,B), suggesting that the responses of the 5 tsRNAs to carbon ion exposure can be reproduced in other kinds of mice. Further, we tried to verify whether the 5 tsRNA biomarkers can be detected in human serum. The serum samples from 6 healthy volunteers (3 males and 3 females) were collected and 200 μL serum of each sample was used to perform RT-qPCR analysis, and the 5 tsRNAs were amplified by the designed primers in Appendix A. The Ct values of the 5 tsRNA were between 25 and 32 when the relative fluorescence unit (RFU) is 500, and the levels of the 5 tsRNAs were similar in serum of males and females (Figure 6C). These results indicate these 5 tsRNA biomarkers can be easily detected in human serum and their background levels are stable, regardless of sex.

## 3. Discussion

The current widely used method for assessment of radiation exposure is physical dosimeter, because it enables us to quickly obtain the information of radiation type and dose rate in environment [1]. However, physical dosimeters have limitations in predicting individual absorbed dose due to the difference of radiation-induced effects on different individuals. Radiation-responsive biomarkers can indicate the individual absorbed dose and provide information about radiation-induced biological effects, but the assay of traditional biomarkers such as chromosome aberration or micronucleus are complex and time-consuming [6,23], which limits the use in special circumstances or massive detection as physical dosimeters did. Over the years, high-throughput approaches were performed for the discovery of new radiation biomarkers in serum or plasma and many radiosensitive circulating molecules have been found in succession. Some proteins in blood such as Interleukin-2 (IL-2), Interleukin-6 (IL-6), Serum amyloid A (SAA), and Insulin-like growth factor binding 3 (IGFBP-3) were reported to display a progressive dose-dependent increase [24,25,26]. However, the studies showed that the level of most circulating proteins does not change significantly after exposure to the low dose of radiation where the ARS or health risks are negligible. Compared to proteins, exRNAs are stable in preservation and extraction [7,27]. In addition, qPCR-based methods for RNA detection are not only with a high sensitivity but also simple and quick for operation. In recent years, miRNAs in blood indicated a big potential of being radiation biomarkers. The levels of miR-200b, miR-150, miR-30a, miR-30c, miR-29a, miR-29b, and miR-320a etc., in serum or plasma were found associated with X-rays or γ-rays exposure [11,12,28]. Our previous studies also identified five serum miRNAs in response to carbon-ion, iron-ion, and X-ray exposure [29]. In fact, circulating exRNAs in blood are enriched in many kinds of sncRNAs. The studies from Wei et al. and Umu et al. showed tsRNAs present a higher level than miRNAs in extracellular vesicles and human serum [10,14], which is conducive to a measurable biomarker. However, the radiation-responsive tsRNAs in blood have not been reported. Since in many scenarios people may be exposed to the low dose of radiation while appropriate biomarkers are lacking [30,31], the research on low dose responsive biomarkers should be highlighted.

Although heavy ions and protons are significant in space radiation, the biomarkers for these radiation types are rarely reported due to scarcity of particle accelerators. In this study, mice were total-body irradiated by the high energy of carbon ions or proton beams generated by HIRFL (Lanzhou, China) and CYCIAE-100 (Beijing, China). We performed RNA sequencing to generate the comprehensive RNA profiles of serum samples after exposure of mice to carbon ions. Lopez et al. reported that a size filtering of 15–40 nt is sufficient to analyze sncRNAs [32]. With a size selection of 14–40 nt reads, we were able to undertake profiling of serum sncRNAs. Most reads in our results were around 30 nt in length (Figure 1A), while the subtype of tsRNAs (tRF-5c and tiRNA) were mainly at 28–32 nt as reported [15]. The reads statistics after mapping them to tRNA and miRNA databases classified over 75% as tsRNAs and around 1% as miRNAs were identified (Figure 1B), suggesting that the level of tsRNAs is significantly higher than miRNAs in serum. Furthermore, two types of tsRNAs (tRF-5c and tiRNA-5) were distinguished in our research, and they were all represented by 5′-end fragments of mature tRNAs. This finding is in agreement with other reports that 5′-end derived tRFs or tiRNAs have a higher abundance in tsRNAs [33,34]. Moreover, serum tsRNAs in our results presented a uniform length distribution in irradiated groups and control group, indicating that they have conserved expression pattern in different individuals.

With screening and validation by RT-qPCR (Figure 2), the levels of 9 tsRNAs showed decrease after carbon ion exposure. Among them, 4 tRF-5c (tRF-Val-AAC-024, tRF-Gln-CTG-018, tRF-Lys-CTT-008, and tRF-Lys-TTT-019) and 1 tiRNA-5 (tiRNA-Glu-TTC-003) were identified as putative biomarkers for carbon ion, X-ray, and proton exposure. By determining the dose responses of tsRNAs, most of the 5 selected tsRNAs showed significant decrease in mice that received 0.1–2 Gy of different radiations (Figure 3). Compared to the protein biomarkers, serum tsRNAs are more sensitive to the low dose of irradiation. Notably, the level decrease of 5 tsRNAs reached a relative plateau after 0.1 or 0.5 Gy, we predict that the decrease trend after exposure to a higher radiation dose (more than 2Gy) might be persisted at the same level as the irradiation dose under 2 Gy. Moreover, we found that the downward trend of the 5 tsRNAs can be detected from 6 h to 7 days, some of them were able to keep the trend reaching 15 or 30 days (Figure 4). These findings indicate that the expression of tsRNA biomarkers will keep the decrease trend at a certain time point within 7 days, but the expression may recover to a normal level after 15 or 30 days. By comparison, serum proteins such as IGFBP-3 and SAA have a significant response to irradiation within 24 h, and most serum miRNAs have a response window under 7 days post-irradiation [24,25,29,35]. In addition, we established the multi-factor model combining the 5 tsRNA biomarkers by multiple logistic regression (Figure 5). This method has been widely used in disease prediction and economics to evaluate the effect of various factors on the outcomes, but it has not been used in predicting radiation exposure [36,37,38]. We observed that the multi-factor models for carbon ions, X-rays, or protons are effective to triage the mice which received different degrees of radiations. Similar to this pattern, the model can be extended to many other biomarkers to form a new superior method for radiation dosimetry.

The tsRNAs have been showed widespread and conserved in a variety of species [15]. We found the level decrease of the 5 tsRNAs after carbon ion exposure also can be reproduced in serum of C57BL/6 and Balb/c mice. Moreover, the 5 tsRNAs were easily detected in human serum and their levels are similar in males and females (Figure 6). These findings indicate that the radiation responses of tsRNA biomarkers obtained from a murine model are available for human beings. However, as this study is limited to male mice, more research on female mice is needed to expand our results to different sexes. In addition, more explorations in other animal models or human beings should be developed before the tsRNA biomarkers could potentially be used as novel radiation biomarkers. Although we discovered the blood tsRNA biomarkers of radiation exposure, the reason that caused the differential expression of tsRNAs remains unclear. The tsRNAs are generated from the specific cleavage of tRNAs by RNases, while RNA modifications within tRNAs play important roles in the determination of the efficiency and specificity of cleavage [39]. Rashad et al. reported that m1A demethylase ALKBH1 can promote tRNA cleavage in a stress-specific manner [40]. Therefore, ionizing radiation may influence the modifications of tRNAs or RNase which causes the level changes of tsRNAs in cells or blood. In recent studies, tsRNAs have been suggested to play a regulatory role by RNAi or binding with ribosome and translation factors [17,18]. They are involved in transcription, translation, ribosome biogenesis, cell proliferation, apoptosis, modulation of the DNA damage response, vesicle-mediated intercellular communication, tumor suppression, and neurological disorders [15,21,41,42,43]. Our results showed that ionizing radiation can affect the level of circulating tsRNAs in the long term. These studies imply that the radiosensitive tsRNAs in blood may chronically affect the process of DNA damage and damage repair, free-radical production, and other functions. However, as the integrated database for prediction of target genes and biological functions of tsRNA is scarce, we are unable to analyze the biological roles of these tsRNA biomarkers by bioinformatics yet. Therefore, follow-up studies in cell models are needed to reveal the underlying mechanism and functions of tsRNAs in radiation responses.

In conclusion, this study firstly reported 5 tsRNAs in serum which positively respond to carbon ion, X-ray, and proton exposure and present a wide time range of response. As the tsRNA is stable and abundant in blood, our results opened the prospect in development of circulating tsRNAs as new biomarkers for the complex radioactive environment. Furthermore, the multi-factor models based on tsRNA biomarkers to predict exposure degree was demonstrated. The assay of tsRNA biomarkers in blood will provide us a new rapid and minimally invasive method for triage or dose assessment within 4 h from blood collection in exposure scenarios, especially for applications in nuclear accidents or space stations.

## 4. Materials and Methods

### 4.1. Mice and Irradiation

Kunming, Balb/c, and C57BL/6 male mice (6–7 weeks old) were purchased from Gansu University of Chinese Medicine (Lanzhou, China) and acclimatized for one week under a 12 h light/dark cycle with standard NIH31 diet (KEAO XIELI Feed Ltd., Beijing, China) and water ad libitum before exposure. During the irradiation, an awake mouse was put into a 50 mL centrifuge tube with an opening, and the beams were vertically downward to the body of mouse at an angle of 90°. The irradiation geometry is as shown in Appendix A. Mice were total-body irradiated by carbon ions, protons, or X-rays. The absorbed dose designed in experiments is described in the result sections. Carbon ions with an energy of 80 MeV/u were generated by the Heavy Ion Research Facility in Lanzhou (HIRFL), Institute of Modern Physics, Chinese Academy of Sciences (Beijing, China) at the dose rates of 0.5 Gy/min. Protons with an energy of 90 MeV/u were generated by 100MeV high intensity proton cyclotron (CYCIAE-100), China Institute of Atomic Energy (Beijing, China) at the dose rates of 0.5 Gy/min. X-rays (225 kV, 13.3 mA) were generated by X-RAD 225 (Precision X-ray, North Branford, CT, USA) at the dose rate of 0.5 Gy/min. Control animals were sham-exposed in the same conditions. All experiments with mice were conducted in accordance with the Guide for Care and Use of Laboratory Animals as adopted and promulgated by the United National Institutes of Health and were approved by the Animals Studies Committee of Institute of Modern Physics, Chinese Academy of Sciences (approval code: No. 2020-012; approval date: 25 June 2020).

### 4.2. Mice Serum Extraction

Mice received general anesthesia by intraperitoneal injection with thiopental, and peripheral blood was collected by heart puncture at the designed time points into RNase-free centrifuge tubes (Kirgen, Shanghai, China). The blood samples in the tubes were stood for 2 h at room temperature. Subsequently, blood samples were centrifuged at 4000 rpm for 10 min at room temperature. Then, the supernatant was separated and centrifuged again at 4000 rpm for 10 min at 4 °C to remove residual blood cells. The last supernatant is a pure serum sample and needs to be stored at −80 °C.

### 4.3. Human Blood Sampling

To verify whether the tRNA derived fragments could be detected in human serum, 1mL veinal blood was collected into RNase-free centrifuge tubes (Kirgen, Shanghai, China) from healthy volunteers (*n* = 6). The cell-free serum was harvested by the same process as above. All volunteers were given the informed consent for the collection of specimens.

### 4.4. Serum RNA Sequencing

Three serum samples in each dose group were mixed together to ensure sufficient volume for sequencing. The total RNA of mice serum was extracted by miRNeasy Serum/Plasma Kit (QIAGEN, Valencia, CA, USA) following the manufacturer’s protocol. The quality and concentration of RNA samples were determined by NanoDrop ND-1000 (Thermo scientific, Waltham, MA, USA). Total RNA samples were pre-treated by rtStar™ tRF&tiRNA Pretreatment Kit (Arraystar, Rockville, MD, USA) to remove some RNA modifications which influence the small RNA-seq library construction. Then, the total RNA of each sample was ligated to 3′ and 5′ small RNA adapters, and cDNA was amplified using Illumina’s proprietary RT primers and amplification primers (Illumina, San Diego, CA, USA). Subsequently, the polyacrylamide gel electrophoresis (PAGE) was used to extract the target cDNAs. The 134–160 bp PCR amplified fragments were extracted and purified from the PAGE gel. The sequencing experiment was performed using Illumina NGS DNA/RNA Sequencing Kit (Illumina, San Diego, CA, USA) following the protocol. The cDNA libraries were diluted to a loading concentration of 1.8pM. Then, diluted libraries were loaded onto a reagent cartridge and carried on Illumina NextSeq 500 system (Illumina, San Diego, CA, USA), according to the manufacturer’s instructions. Finally, the raw data generated from Illumina NextSeq 500 that pass the Illumina chastity filter were used for the following analysis. The overview flowchart of RNA-seq is shown in Appendix A. The reads were mapped with GtRNAdb: Genomic tRNA Database (http://gtrnadb.ucsc.edu/ (accessed on 10 December 2021). The trimmed reads were aligned allowing for 1 mismatch only to the mature tRNA sequences, then reads that did not map were aligned allowing for 1 mismatch to precursor tRNA sequences with bowtie software. The remaining reads were aligned allowing for only 1 mismatch to miRNA reference sequences with miRDeep 2.0 (Berlin, Germany). The workflow of data analysis is shown in Appendix A. The abundance of sncRNAs was evaluated by the counts per million of total aligned reads (CPM). The sncRNA with CPM of more than 20 was selected for further analysis. The relative expression of tsRNAs compared to the control group was calculated and the differentially expressed tsRNAs were selected out.

### 4.5. Quantitative RT-PCR

The tsRNAs selected out from the expression profiles were validated by quantitative real-time polymerase chain reaction (RT-qPCR) to observe their dose- and time-responses. Total RNA in serum was extracted using miRNeasy Serum/Plasma Kit (QIAGEN, Valencia, CA, USA) following the manufacturer’s protocol. Before RNA extraction, the cel-miR-39 (QIAGEN, Valencia, CA, USA) from the kit was added to serum samples for normalization. Total RNA reverse transcription was carried out using miRNA First Strand cDNA Synthesis Kit (Sangon, Shanghai, China) according to the protocol of the manufacturer. Template amplification was performed by GoTaq SYBR Green qPCR Master Mix (Promega, Madison, WI, USA). The tsRNAs specific custom primers were designed and synthesized by Sangon Biotech (Shanghai, China). The PCR procedure was carried out with a CFX96 Touch Real-Time PCR system (Bio-Rad, Hercules, CA, USA). The reaction program is as following: initiation for 10 min at 95 °C, followed by 45 thermal cycles each at 95 °C for 10 s and at 60 °C for 20 s, and at 70 °C for 10 s.

### 4.6. Statistical Analysis

Data of RT-qPCR were analyzed using 2^−^^ΔΔCt^ method as reported [44]. Data normalization in group was performed using exogenous cel-miR-39. All groups contain at least three independent biological repetitions and all data were presented as the mean ± standard error. One-way analysis of variance was used to determine the differences in multiple groups, and Student’s *t*-test was used to determine the difference between treatment groups and control group. Statistical analysis was performed using SPSS version 18.0 software (IBM Corp, Armonk, New York, NY, USA). Multiple linear regression analysis was also processed using SPSS 18.0. Receiver-operating characteristic (ROC) curves analysis was processed and graphs were generated by MedCalc version 15.0 software (Ostend, Belgium).

## Figures and Tables

**Figure 1 ijms-22-13476-f001:**
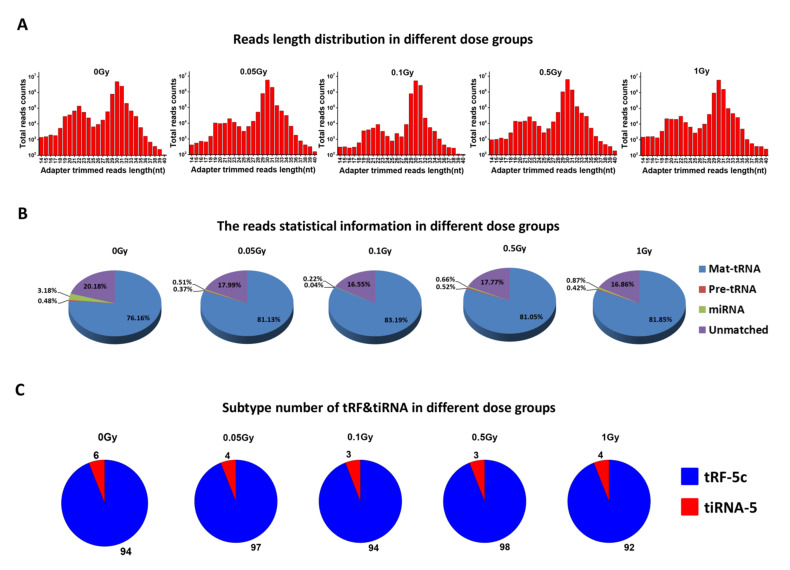
The overview of length distribution and classification of tsRNAs in serum of mice after carbon ion exposure. (**A**) Total read counts against the lengths of 14–40 nt RNA reads in different dose groups. (**B**) The statistical information of mapped reads in different dose groups. (**C**) Subtypes of tsRNAs in different dose groups.

**Figure 2 ijms-22-13476-f002:**
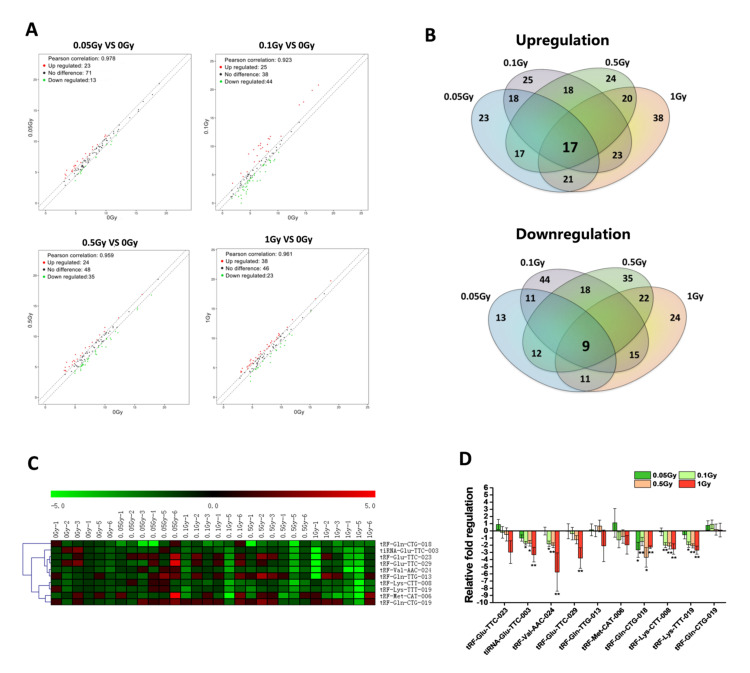
Screening and validation of differentially expressed serum tsRNAs after carbon ion irradiation. (**A**) The scatter plots showing the relative expression of serum tsRNAs in mice of exposure to 0.05, 0.1, 0.5, and 1 Gy of carbon ions (the red dots above the top line or the green dots below the bottom line indicate more than 1.5 times fold change between the two compared groups). (**B**) Venn diagram showing the differentially expressed tsRNAs in different dose groups. (**C**) The relative expression changes of the 10 selected tsRNAs were validated by RT-qPCR and the results are shown by hierarchical clustering heat map (red indicates upregulation and green indicates downregulation). (**D**) The fold regulation of the 10 selected tsRNAs in response to different doses of carbon ion irradiation. * *p* < 0.05, ** *p* < 0.01, compared with control group (0 Gy).

**Figure 3 ijms-22-13476-f003:**
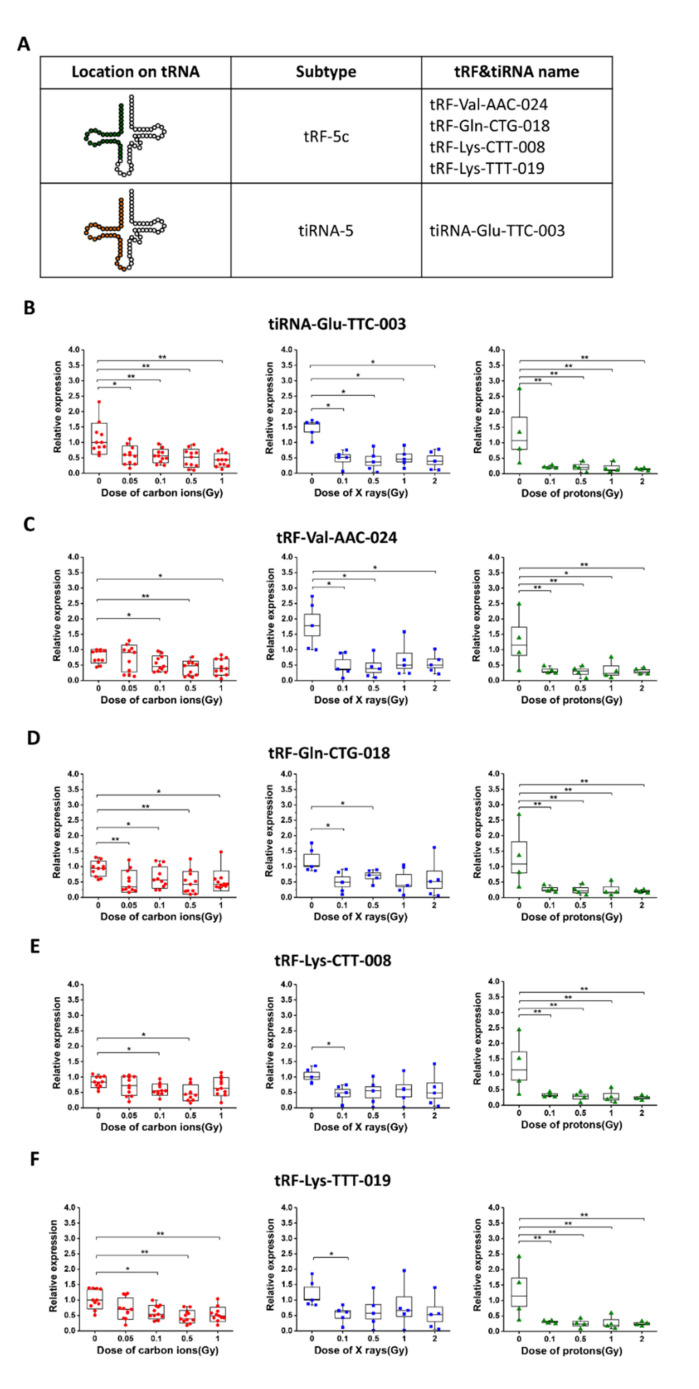
The dose responses of five selected tsRNAs after exposure of mice to carbon ions, X-rays, and protons. (**A**) Classification of the five selected tsRNAs (tiRNA-Glu-TTC-003, tRF-Val-AAC-024, tRF-Gln-CTG-018, tRF-Lys-CTT-008, and tRF-Lys-TTT-019) and their location on the structure of tRNA. (**B**–**F**) The relative expression of the five tsRNAs in response to different dose exposure of carbon ions (*n* = 11/dose), X-rays (*n* = 5/dose), or protons (*n* = 4/dose). Each dot represents the relative expression of tRF or tiRNA in each individual, * *p* < 0.05, ** *p* < 0.01, compared with control group (0 Gy).

**Figure 4 ijms-22-13476-f004:**
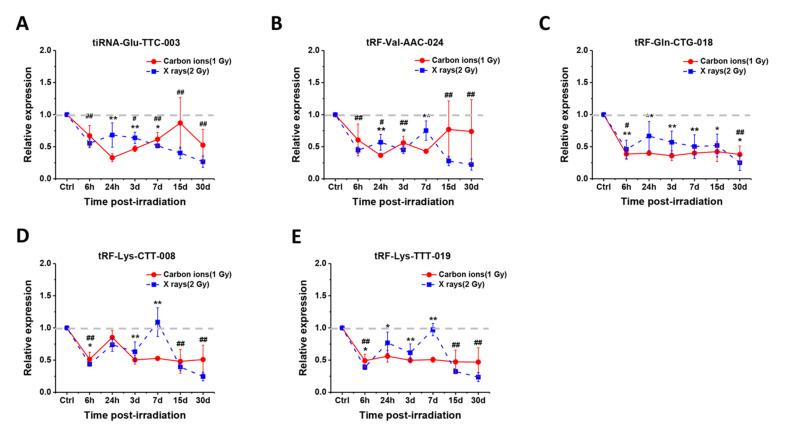
Temporal changes of the five selected tsRNAs in serum of mice after carbon ion or X-ray exposure. (**A**–**E**) Relative expression of the five selected tsRNAs (tiRNA-Glu-TTC-003, tRF-Val-AAC-024, tRF-Gln-CTG-018, tRF-Lys-CTT-008, and tRF-Lys-TTT-019) from 6 h to 30 days (d) after exposure of mice to 1 Gy of carbon ions or 2 Gy of X-rays. The baseline of the expression is indicated by a grey dotted line. In groups of carbon ion irradiation, * *p* < 0.05, ** *p* < 0.01; in groups of X-ray irradiation, # *p* < 0.05, ## *p* < 0.01; compared with control group (0 Gy).

**Figure 5 ijms-22-13476-f005:**
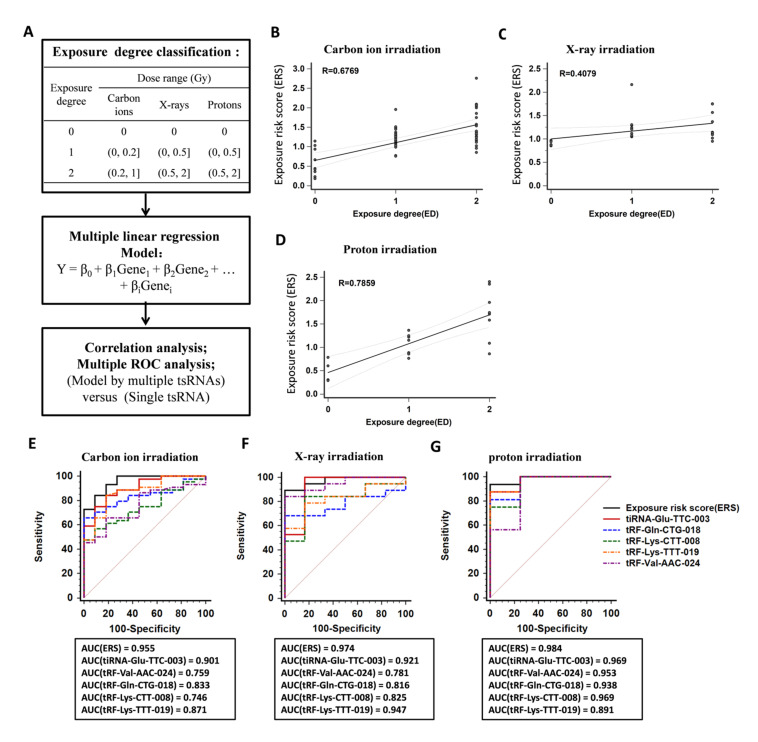
Establishment of a model based on multiple tsRNA biomarkers to indicate the degree of carbon ion, X-ray, or proton exposure. (**A**) A flow chart showing the establishment of the multi-factor models. (**B**–**D**) The relationship between exposure degree (ED) and exposure risk score (ERS) was predicted by the models from dose–response data of carbon ion, X-ray, and proton exposure. (**E**–**G**) Comparisons of ROC curves among the model and single tsRNA in triage of irradiated mice (AUC: area under ROC curve; Sensitivity: the percentage of individuals with irradiation who are correctly identified by the test; 100-Specificity: the percentage of individuals without irradiation who are correctly excluded by the test).

**Figure 6 ijms-22-13476-f006:**
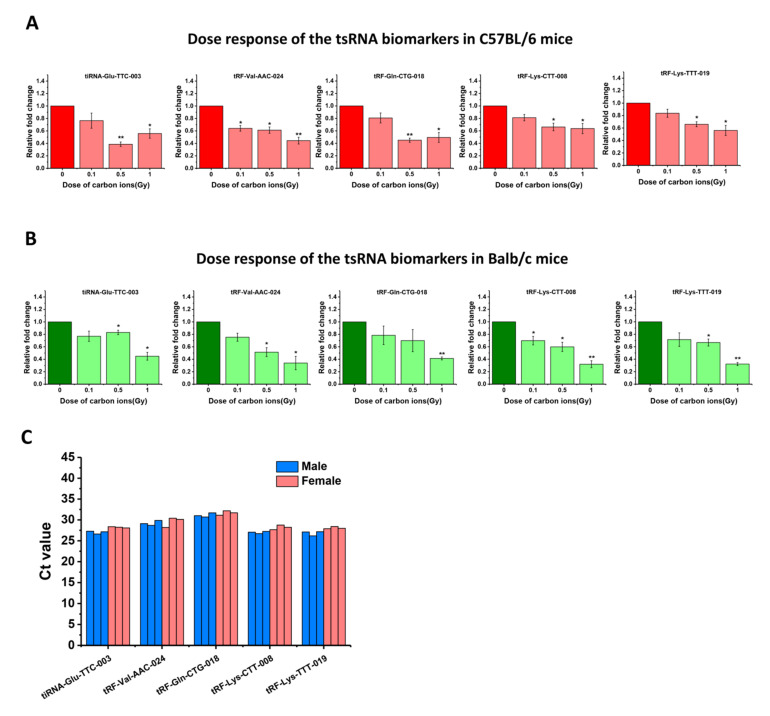
The radiation responses of the five tsRNA biomarkers in different mice species and their expression levels in human serum. (**A**) The relative expression of the five tsRNAs responding to carbon ion irradiation in C57BL/6 mice. (**B**) The relative expression of the five tsRNAs responding to carbon ion irradiation in Balb/c mice. * *p* < 0.05, ** *p* < 0.01, compared with control group (0 Gy). (**C**) The expression of five tsRNA biomarkers in serum sample of six healthy humans (200 μL/sample) was detected by RT-qPCR. The Ct value of the five tsRNAs in each sample is as shown.

## Data Availability

The data presented in this study are available on request from the corresponding author.

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
