# Peer review of "Circulating tRNA-Derived Small RNAs as Novel Radiation Biomarkers of Heavy Ion, Proton and X-ray Exposure"

_ijms, 2021, doi:10.3390/ijms222413476_

Round 1

Reviewer 1 Report

Dear Authors,

Though you presented all the experimental part and result in a very descriptive way to prove that the circulating tsRNAs can serve as new minimally invasive biomarkers for rapid triage or dose assessment in exposure condition. I have some query regarding the dose and gender specificity:

  1. As lots of studies are focused on gender based radiation sensitivity. Did you perform any gender based study to identify circulating tsRNA expression in male and female? It is better to perform some gender based tsRNA expression in male and female mice serum after exposure to radiation.
  2. What will happen to the tsRNA expression if radiation dose is higher than the radiation dose used in this study? Number of tsRNA expression will increase or decrease, please give some data on this issue.
  3. What is the survival rate of the mice after giving the radiation?
  4. What will be the expression of tsRNA in those mice which survived after a certain time point? Please describe or add some data to make your manuscript more impactful.

Thank you.

Reviewer 2 Report

Review International Journal of Molecular Sciences: Circulating tRNA-derived small RNAs as novel radiation biomarkers of heavy-ion, proton, and X-ray exposure.

This original research manuscript identifies that three different radiation exposure sources downregulate tRNA-derived small RNAs. The authors corroborated data that mature tRNAs, specifically tRF-5cs, were the most abundant subset in Kunming mice. Although changes in miRNA levels due to radiation exposure were observed previously, it was not known whether tRNA-derived small RNAs levels were impacted in response to radiation. The authors treated the mice with different radiation doses and sources and found that many changes (upregulation/downregulation) of these tRNAs occurred in the Kunming mice. Five downregulated tRNAs were selected, and their levels in response to carbon ions, x-rays, and proton exposure were measured. For the most part, all tRNAs were downregulated in response to radiation exposure within 24 hours and found that utilizing this technique was optimal for only about 6-7 days after exposure. To assess specificity, the authors recapitulated these experiments in two other strains of mice. Furthermore, identified that these five tRNAs were present in human blood samples. With their data, the authors generated a model using the tRNA biomarkers as indicators of the degree of radiation exposure. The pilot data are robust and adds to the growing body of literature in the field of radiation biodosimetry.

Strengths:

Overall, this paper is well-written and very interesting. The author’s science was robust, testing three different mouse strains, three different exposure types, four different exposures, and collected samples at different time points after exposure. In addition to identifying tRNAs as novel biomarkers for radiation exposure, the authors took this a step further, constructing a model to calculate the dose of radiation exposure. Another strength is that the authors showed that the biomarkers are most adequately utilized within the first 6-7 days of exposure. This is a great start for establishing tRNAs as biomarkers for radiation exposure.

Weaknesses:

The major weakness encompasses the connection to human biology the authors attempted to make when analyzing human blood samples for the composition of basal tRNA levels. The authors validated that the tRNAs found in the three mouse strains were present in human females and males similarly. However, the obvious question, "what happens after radiation exposure" was not answered. This is hard to answer for higher radiation doses. However, is it possible to obtain blood samples post-radiation of cancer patients? Also, maybe recapitulating in treated and non-treated cells to assess the role of radiation on human tRNAs could also work.

Minor Weaknesses:

  • Abstract: Although the authors state that the tsRNA has the potential for rapid triage and dose assessment in an exposure scenario, the reviewer requests that the authors define what their definition of ‘rapid’ is. Requirement for a triage poc device for biodosimetry is 30 min from sample to answer and 4 h for a definitive dose estimation (PMID: 27590469).
  • This reviewer is concerned that the authors have not clearly identified the need for assessing radiation dose at low doses. Further, it is inaccurate to state that the classic techniques such as the DCA does not capture exposure at low doses of 0.5 Gy while literature clearly indicates otherwise. Further, is there a need for this assessment for radiotherapy subjects where carefully calibrated doses are administered to patients and the doses are known in advance? Also, authors are requested to address their meaning of ‘rapid’ since a PCR run takes a minimum of several hours.
  • Results: Is the RBE for Carbon beam, protons and x-rays comparable at the dose rates utilized?
  • Figure 1A: Some graphs are hard to read because the font is too small.
  • Page 2, Line 75: No need to add 0 as a radiation timepoint. It seems a little redundant. I think it is a given that you would be comparing to basal levels. Maybe something like non-irradiated mice were compared to mice irradiated with 0.05, 0.1, 0.5, and 1 Gy.
  • Page 2, Line 75: 1Gy was the maximum amount of radiation given to the mice. Up to 1 Gy would induce subclinical Acute Radiation Syndrome. This is good if you want to evaluate low-dose radiation exposure, but what about higher exposures? It would be important to test higher exposures, no greater than 10Gy, to make your work relevant to industrial accidents or nuclear attacks.
  • Page 2, Line 75, Remove accidents at the end of terrorist nuclear accidents. These are not accidents. They are intentional and malicious.
  • Page 2, Line 76: More details on how this was done in your results session to flow better, and so we know how you performed the assay. (details on procedure within the text may clarify this: You focus so much on these RNAs being present in extravesicular bodies, which are hard to purify, and people may have questions about how you are enhancing these vesicles that contain the tRNAs. I would suggest you de-emphasize more about their location).
  • Fig 2, panel B. The number of upregulated tsRNA are much higher than the down-regulated genes. Can the authors justify why they selected only the down-regulated tsRNAs to form their panel? Why not the upregulated genes? Or a combination of up- and down-regulated tsRNAs?
  • Fig 3. The downward trends in these genes for all three types of radiation does not appear to be radiation-dose dependent. How does this inform the definitive dose aspect of the proposed use?
  • Fig 4. It would be tremendously useful to have a baseline bar for Carbon and x-rays so that the readers can follow the trends longitudinally. For instance, so these tsRNA change over time?
  • Fig 5, E-G. In addition to the combined 5 tsRNA and single gene ROC analysis, did the authors attempt other permutations and combinations ranging from 2-4 gene combinations?
  • Fig 6. The reviewer is extremely pleased to see human data for these biomarkers. The authors are cautioned to consider early on that the response of these trends in humans be considered (from radiotherapy patients etc ) to ensure that the response to radiation is similar in both mice and men.
  • Page 10, line 213: replace gender with sex, gender is a social construct of female versus male characteristics versus sex refers to the biology
  • Page 10, line 211: remove different, different is not needed
  • Discussion, pg 11. Ln 234-235. Again the reviewer requests that the authors clarify their focus on very low doses of radiation, where health risks are negligible.
  • Ln 251. Please change ‘scarce’ to ‘scarcity’.
  • Ln 278. Please clarify what you mean by ‘prediction models’. Are you predicting an outcome following irradiation? Or do you mean that you are able to accurately estimate the radiation dose that the individual has been exposed to?
  • 1 M&M Please describe how the mice were irradiated. Were they anesthetized and irradiated? Where they placed in jigs (add dimensions and material of the jig)? Was an in-run radiation measurement conducted to ensure accuracy of administered dose?
  • Page 12, Line 293: a better discussion of the role of these tRNAs is needed.

Reviewer 3 Report

This manuscript proposes a panel of five circulating tRNA-derived small RNAs as novel biomarkers for exposure of mammals to ionizing radiation. Since tRNA-derived RNAs have recently been shown to be involved in many different regulatory processes, their involvement in radiation response is per sè very interesting. Moreover, the proposed biomarkers have two very appealing features: they are modulated by both electromagnetic and particle radiation and the modulation persists for a long time. These features make them very promisful even if they were not tested in human. For these reasons the manuscript is interesting for the general audience and for the radiobiologists in particular and deserves publication in IJMS.

On the other hand, some changes need to be done in order to improve text clarity and results presentation.

Major points:

1 – In the discussion the Authors state: “In terms of sensitivity to low dose of irradiation, the levels of 5 selected tsRNAs significantly decreased in mice that received as low as 0.05 Gy or 0.1 Gy of irradiation” and “In conclusion, this study firstly reported 5 tsRNAs in serum which positively respond to low dose of carbon ions, X-ray and proton exposure.” This statements are not supported by data. Indeed, in C57Bl/6 and BalB/c, only 1/5 of the tRNA biomarkers significantly responds to carbon ions doses of 0.1 Gy and in Kunming mice only 3/5 significantly respond to 0.05 Gy of carbon ions, while 0.05 Gy of X rays were not tested. 0.1 Gy of carbon ions or protons cannot be defined as “low dose”. I think that a high sensitivity is not an advantage of these biomarkers while responsiveness to different quality radiation and wide time range of response are.

2 – The protocols should be better explained. As I understand RNAseq was made on pools of three individual serums for each group. Then tsRNAs modulated more than 1.5 times were selected for validation with no statistical filtering but with intensity filtering (how many reads/million reads?). Then for PCR “all groups were independently repeated three times at least…” what do you mean? Three independent pools? Three independent individuals? Three independent technical replicas of the same pool? How about fig.3, panels B-F? Here it looks as you did many different biological replicas for each group but n is not indicated.

Minor points

  • In the introduction the Authors should insert some more recent literature on tsRNA putative regulatory role because this field is going fast (i.e. Polacek and Ivanov “The regulatory world of tRNA fragments beyond canonical tRNA biology” RNA BIOLOGY 2020, VOL. 17, NO. 8, 1057–1059 and some references therein).
  • Table 1 should be transferred in the supplementary materials and edited since the sequences are not correctly written (write 5’-NNNNN-3’ without interrupting the sequence).
  • 5 B-C-D: name the Y-axis (ERS) and explain in the legend what is represented by the X-axis in panels E-F-G. The meaning of ROC curves should also briefly explained in the text for those who are not familiar with them.
  • 6 panel C is not meaningful and should be removed. Panel D is sufficient to show that the putative biomarkers are reproducibly detectable in human, provide that the Authors state the volume of the serum used in the materials and methods or in the figure legend. The indication of the asterisks’ meaning should be transferred at the end of panel A and panel B description.
  • Page 2, line 85: delete ,besides, line 88 …while only 3-6 % of the fragments
  • Page 3, line 96 …changes higher than… line 104 …were shown by fold change histogram ..….105 We found that the level of the 10 most modulated tsRNAs decreased….106 in at least three dose groups
  • Page 9, line 201 …in different mouse strains, C57BL/6 and Balb/c mice…
  • Page 10, line 206 …in other mouse strains. Line 207 ..biomarkers
  • Page 11, line 235 ……does not change line 240 …..were found associated  line 246 Since in many scenarios   line 248 …the research    line 252 …energy carbon ions  or proton beams generated by.. line 255 ….reported that  a size filtering… line 257 …were around 30 nt in length line 259 …mapping them to tRNA and MiRNA databases classified over 75% as tsRNA and around 1% as miRNAs   line 260 …suggesting that    line 262 …they were all represented by 5’-end fragments of mature tRNAs  line 270 ….as putative biomarkers…
  • Page 12, line 288 …were easily detected in human serum line 291 …could potentially be used in…

Round 2

Reviewer 1 Report

Dear Authors,

Thank you for incorporating the suggestion in the revised manuscript.